# Cerebellar Differences after Rehabilitation in Children with Developmental Coordination Disorder

**DOI:** 10.3390/brainsci12070856

**Published:** 2022-06-29

**Authors:** Kamaldeep K. Gill, Donna Lang, Jill G. Zwicker

**Affiliations:** 1Graduate Programs in Rehabilitation Sciences, University of British Columbia, Vancouver, BC V6T 2B5, Canada; k.gill@alumni.ubc.ca; 2Brain, Behaviour, & Development Theme, BC Children’s Hospital Research Institute, Vancouver, BC V6H 3V4, Canada; donna.lang@ubc.ca; 3Department of Radiology, University of British Columbia, Vancouver, BC V5Z 1M9, Canada; 4Department of Occupational Science & Occupational Therapy, University of British Columbia, Vancouver, BC V6T 2B5, Canada; 5Department of Pediatrics, University of British Columbia, Vancouver, BC V6H 0B3, Canada

**Keywords:** developmental coordination disorder, motor skills disorder, rehabilitation, CO-OP, voxel-based morphometry, cerebellum, neuroplasticity

## Abstract

Developmental coordination disorder (DCD) affects a child’s ability to learn motor skills. Cognitive Orientation to daily Occupational Performance (CO-OP) is one of the recommended treatments to help achieve functional motor goals. The purpose of this study was to determine if CO-OP intervention induces functional improvements and structural changes in the cerebellum of children with DCD. Using a randomized waitlist-controlled trial, we investigated the effects of CO-OP intervention on cerebellar volume in 47 children with DCD (8–12 years old). Outcome measures included the Canadian Occupational Performance Measure, Performance Quality Rating Scale (PQRS), and Bruininks–Oseretsky Test of Motor Proficiency-2. The SUIT toolbox was used to carry out voxel-based morphometry using T1-weighted MRI scans. Children with DCD showed improved motor outcomes and increased gray matter volume in the brainstem, right crus II, bilateral lobules VIIIb, and left lobule IX following CO-OP. Significant associations were found between PQRS scores and regional gray matter changes in the brainstem, right crus II, right lobule VIIb, right and left lobule VIIIb, and vermis IX. Given the improved motor and brain outcomes with CO-OP, it is recommended that children with DCD be referred for this rehabilitation intervention.

## 1. Introduction

Developmental coordination disorder (DCD), a neurodevelopmental disorder affecting motor coordination and motor learning, significantly interferes with a child’s ability to execute activities of daily living (i.e., buttoning clothes and tying shoelaces) and/or academic achievement [1]. DCD is a major contributor to childhood motor impairment, affecting up to 5–6% of school-aged children [1]. DCD is often associated with other neurodevelopmental disorders, with 50% of DCD children being diagnosed with attention deficit hyperactivity disorder (ADHD) (and vice versa) [2]. Without appropriate intervention during childhood, up to 75% of DCD children will continue to experience impairments well into adolescence and adulthood [3].

The etiology of DCD is unclear. Differences in brain structure, function, and development have been identified [4]. Several brain regions have been implicated, particularly the basal ganglia, parietal lobes, medial orbitofrontal cortex, dorsolateral prefrontal cortex, and cerebellum [5]. Cerebellar deficits may be a key contributor to the emergence of DCD, given its role in motor coordination, motor learning, and executive functioning [5].

Children with DCD need support to develop problem-solving and self-regulatory skills to address motor performance difficulties. It is posited that motor difficulties in children with DCD may be partly due to impairments in self-regulation (i.e., monitoring performance) and emotional regulation (e.g., sustaining motivation and attentional regulation) [6]. The Cognitive Orientation to daily Occupational Performance (CO-OP) intervention aims to support the development of cognitive and self-regulatory strategies to address motor performance difficulties [7,8]. CO-OP is a 10-week, client-centered, direct intervention focusing on (a) skill acquisition, (b) cognitive strategy development and strategy application, and (c) generalization and transfer of skills to other daily tasks [7]. In CO-OP, children set their own therapy goals and are guided to discover and develop their own cognitive strategies to achieve their goals through the global strategy *Goal-Plan-Do-Check* [7]. CO-OP is considered to be an effective approach for learning, maintaining, and transferring strategies related to the performance of fine and gross motor skills [7].

While CO-OP has been effective in improving motor outcomes and functional goals in individuals with DCD, potential associated cerebellar neuroplasticity has been largely unexplored. The aim of this study was to determine (i) if CO-OP intervention was associated with cerebellar neuroplasticity and (ii) if there was an association between potential cerebellar structural changes and motor improvements.

## 2. Materials and Methods

### 2.1. Study Design

This study is part of a larger randomized control trial investigating brain structure and function in children with DCD, children with DCD + ADHD, and typically developing children (ClinicalTrials.gov ID: NCT02597751). The current pre-test/post-test investigation examined potential structural differences in the cerebellum following CO-OP intervention. The sample size calculation was calculated for the larger study, resulting in a sample of 25 participants per group for 80% power, standard deviation of 2.5, and *p*-level of 0.05, to detect a significant difference in the main motor outcome measure (Canadian Occupational Performance Measure (COPM)). We aimed to recruit a target sample size of 30 per group based on the power calculation for neuroimaging measures.

### 2.2. Participants

A sample of 80 children (8–12 years) with DCD was recruited from the Sunny Hill Health Centre DCD Clinic, the BC Children’s Hospital ADHD Clinic, or the Greater Vancouver area community. Inclusion criteria were based on DSM-5 guidelines [1] and international clinical practice guidelines [6] for DCD as follows: (a) score ≤16th percentile on the Movement Assessment Battery for Children—2nd edition (MABC-2) [9]; (b) score in the suspected or indicative range on the DCD Questionnaire (DCDQ) [10]; (c) parent-reported motor difficulties from a young age; and (d) no other medical condition that could explain motor difficulties as per parent reports, clinical reports, and/or medical examination. Children with a co-occurring ADHD diagnosis, as per parent report, were included in the study as DCD children are more likely than typically developing children to have attentional difficulties [2]. Exclusionary criteria included being born preterm (gestation < 37 weeks) or having a diagnosis of autism spectrum disorder.

### 2.3. Procedures

Study approval was provided by the Children’s and Women’s/University of British Columbia Clinical Research Ethics Board. Parents/legal guardians provided informed written consent and children provided assent. Prior to enrollment, all participants were administered the MABC-2 to quantify the level of motor impairment and to determine if participants met the inclusion criteria. Additionally, the Conners 3 ADHD Index parent form was used to determine and assess ADHD symptoms [11].

An MRI safety screening and MRI simulator session was carried out to familiarize every child with the MRI environment and to alleviate associated anxiety. In the larger randomized controlled trial, children completed MRI sessions at baseline, at three months, and at six months. Children in the treatment group received intervention between the first and second MRI. Children in the waitlist group received intervention between the second and third scans. To obtain sufficient power to examine differences in cerebellar gray matter volume before and after intervention, we combined Scan 1 of the treatment group and Scan 2 of the waitlist group to create the “pre-intervention” group, and combined Scan 2 of the treatment group and Scan 3 of the waitlist group to form the “post-intervention” group. The MRI technicians were unaware of group assignment but were aware of which scan was being conducted due to the nomenclature of how scans are labelled. Before and after intervention, an independent occupational therapist administered the motor outcome measures COPM [12], Performance Quality Rating Scale (PQRS) [13], and Bruininks–Oseretsky Test of Motor Proficiency-2 (BOT-2) [14].

### 2.4. Outcome Measures

The COPM is a well-validated outcome measure of self-perceived motor performance and satisfaction of each of the child’s self-chosen goals [12]. Motor performance and satisfaction are rated on a scale from 1 to 10, with a higher score indicating increased performance and satisfaction with their functional motor goals [12]. A change of two or more points on the COPM is considered to be clinically meaningful [12].

The PQRS is an observational, video-based tool that measures performance on child-chosen activities [13]. Children were recorded performing their chosen goals before and after CO-OP intervention. The recordings were scored by an occupational therapist not involved in delivering the intervention and who was blinded to the pre-/post-assessment sessions. Ratings of 1 (“can’t do the skill at all”) to 10 (“does the skill very well”) were used to score movement quality on the first and last session [13]. An increase of three points is considered clinically meaningful [13]. The PQRS has moderate to substantial inter-rater reliability, excellent test–retest reliability, and a good internal responsiveness, as evidenced by large effect sizes for children with DCD [13].

The BOT-2 is a well-validated, standardized, discriminative, norm-referenced assessment that measures motor performance [15,16]. We used the short form of BOT-2, which consists of one or two items from eight areas of motor functioning [14]. The BOT-2 percentile scores were used for analysis.

### 2.5. Intervention

The CO-OP intervention was administered by registered occupational therapists trained in the research protocol. The children were seen for one-hour sessions once weekly for 10 weeks as per published protocol [7]. During the first session, the parents received training to apply CO-OP strategies at home, between treatment sessions. The parents were given a weekly logbook at the first session to track the amount of time the child practiced each goal at home over the 10 weeks.

### 2.6. Data Collection

MRI data were acquired on a 3-Tesla General-Electric Discovery MR750 MRI scanner using a 32-channel head coil. The parameters for the T1-weighted images were acquired as follows: 3D spoiled gradient recalled echo; echo time = 30 ms, repetition time = 3000 ms, field of view = 256, matrix size = 256 × 256, flip angle = 12°, slice number = 256, slice thickness = 1 mm, interleaved no gaps (voxel size 0.9375 × 0.9375 × 1 mm). All scans were visually inspected for motion-related artifacts [17]. Scans with incomplete coverage, significant scanner or motion artifacts, or poor gray/white matter differentiation were excluded [17]. An ordinal score was given to each image based on motion artifacts and image quality (pass, questionable, or fail) using standardized methodology [18]. Only scans that were ranked as a pass were retained for analysis.

The images were processed using voxel-based morphometry (VBM) [19]. The Spatially Unbiased Infratentorial (SUIT) toolbox [20] was used for cerebellar VBM through SPM12 in MATLAB 2016a (The MathWorks Inc. Natick, MA, USA). For each participant, the toolbox Isolate function was used to create a cerebellum mask and to generate gray and white matter segmentation maps. Segmentation maps were then normalized to the SUIT template via non-linear DARTEL normalization. The resulting gray matter image was refitted into the SUIT atlas space and scaled by the Jacobian determinant. The images were then smoothed using a 4 mm full width at half maximum (FWHM) Gaussian kernel. All images were inspected for quality at each step. The cerebellum was segmented into 28 gray matter regions of interest, including 10 bilateral lobules (I–X right and I–X left) and the vermis lobules VI–X [20]. The SUIT atlas combines lobules I–IV into one measurement; lobule VII is divided into VIIa, VIIb, Crus I, and Crus II; and lobule VIII is divided into VIIIa and VIIIb [20].

### 2.7. Statistical Analysis

IBM SPSS Statistics for Mac, Version 25.0 (Armonk, New York, NY, USA) was used to summarize the group age, Conners 3 ADHD Index percentile scores, MABC-2 scores, and total intercranial volume (TIV). The Wilcoxon signed-rank test was used to compare the effects of CO-OP on COPM performance and satisfaction scores, PQRS total scores, and BOT-2 percentile ranks. Bonferroni correction was used to correct for multiple comparisons, and the alpha level was set at 0.05.

Pre- and post-intervention volume comparisons were performed using Permutation Analysis of Linear Models (PALM) [21] in the FMRIB Software Library (Oxford, UK). This analysis used threshold-free cluster enhancement (TFCE), a voxel-wise statistical method in which each voxel’s value represents the cluster-like spatial support in accordance with the spatial neighborhood information [21]. A paired *t*-test, with 5000 permutations, whole and within exchangeability blocks, was used to explore group differences in volumes. Additionally, an independent *t*-test was used to compare the treatment and waitlist groups at pre-intervention in order to determine if there was an influence of cerebellar maturation between time points. To assess the association between the BOT-2 and PQRS scores and gray matter volume, regression analysis was used with 5000 permutations. COPM was not included as one of the variables in the regression analysis due to high collinearity with PQRS and BOT-2 measures. There was no correlation between PQRS and BOT-2 measures. TIV was demeaned and entered as a nuisance variable, as it was considered a covariate in this analysis. We used TFCE, family-wise error correction, and a cluster threshold of 25 voxels. The alpha level was set to 0.05.

## 3. Results

### 3.1. Participant Characteristics

The demographics and behavioural characteristics are shown in Table 1. Our final sample included 47 participants (Figure 1), of which 41 (87%) had clinically significant attentional difficulties, as indicated by a score of 70 or greater on the Conners 3 ADHD Index. This sample included 37 males (79%) and 10 females and is representative of typical clinical samples of children with DCD [1].

### 3.2. Motor Outcomes

COPM performance and satisfaction scores, PQRS ratings, and BOT-2 percentile scores were compared before and after CO-OP intervention. Participants showed significant improvements in their motor goals (COPM), movement quality (PQRS), and motor skills (BOT-2), with correction for multiple comparisons (all *p*-values < 0.001) (Table 2).

### 3.3. Cerebellar Gray Matter Volume

#### 3.3.1. Intervention-Induced Gray Matter Changes

First, pre- and post-intervention regional gray matter volumes were compared. Children with DCD showed increased gray matter in the brainstem, right crus II, bilateral lobule VIIIb, and left lobule IX following CO-OP intervention (Table 3; Figure 2). No cerebellar regions had greater gray matter pre-intervention compared to post-intervention. Next, to determine if cerebellar changes could be a result of maturation over the 12 weeks, we compared gray matter volume between the treatment (Scan 1) and waitlist (Scan 2) groups before intervention. We found no significant differences in gray matter volume between groups (*p* > 0.05). To ensure that intervention effects were not due to differences in volume at baseline, we compared total cerebellar gray matter volume, cerebellar white matter volume, and total cerebellar volume between the treatment and waitlist groups at baseline and found no significant differences (*p* > 0.05).

#### 3.3.2. Relationship of Motor Outcomes to Regional Cerebellar Gray Matter Changes

We examined the association of BOT-2 and PQRS scores with changes in regional gray matter volume. No association between overall motor performance on the BOT-2 percentile scores and regional gray matter changes was observed (*p* > 0.05; cluster size <25). There was a significant positive association between movement quality on motor goals (PQRS scores) and regional gray matter changes. Higher PQRS scores predicted increased gray matter volume following CO-OP intervention in the following regions: brainstem, right crus II, right lobule VIIb, right and left lobule VIIIb, and vermis IX (Table 4; Figure 3).

## 4. Discussion

We investigated the effects of CO-OP intervention on regional gray matter changes in the cerebellum, and the relationship between cerebellar gray matter changes and motor outcomes. To our knowledge, this is the first study to exclusively examine structural cerebellar differences following CO-OP intervention in children with DCD. Post-CO-OP intervention, we observed that children with DCD had greater regional gray matter volume in the brainstem and cerebellum, and a corollary positive association with movement quality improvements. Interestingly, many of the regions where we observed increased gray matter volume after intervention were similar to regions we previously reported as beginning smaller in children with DCD compared to typically developing children [22] (i.e., the brainstem, right/left crus I, right crus II, left VI, right VIIb, and right VIIIa lobules).

The cerebellum likely has a unique role in motor adaptation, which involves the modification of learned motor actions in response to contextual changes [23]. Motor actions are modified by updating internal models of movement located within the cerebellum based on the error signal created during motor learning [24]. Children with DCD have difficulties with motor adaptation involving the modification of learned motor action [4]. Typically developing children learn motor skills either implicitly or explicitly by observing and imitating other children and adults or by trial and error using verbal-cognitive strategies [4]. An important factor in motor learning is the inherent ability of typically developing children to monitor their own performance, to detect possible errors, and to identify possible sources of these errors [4]. Children with DCD appear to learn using different strategies and brain regions to improve motor performance and motor learning [25]. It is posited that children with DCD need support in developing problem-solving and self-regulatory skills for addressing motor performance difficulties [26].

The current study indicates that with CO-OP intervention, children with DCD can develop problem solving skills and use compensatory strategies (explicit motor learning) to circumvent deficits in forming or updating internal models of movement (implicit motor learning) in the cerebellum [8]. Similarly, individuals with cerebellar degeneration were able to employ explicit learning strategies in the absence of implicit adaptation for motor adaptation and learning throughout training [27]. Based on the principles of motor learning and neuroplasticity, we hypothesize that children with DCD show improved motor skills and relatively permanent change in function due to the task-specific and cognitive nature of CO-OP intervention.

The ability to use and achieve functional motor goals as a part of CO-OP intervention is reflected in the gray matter changes seen in the brainstem, right and left lobule VIIIb, and left lobule IX. These regions play an important role in sensorimotor functioning and visual guidance of movement [28]. Additionally, we observed that cognitive strategies positively modified the right crus II, a key region involved in higher order cognitive functioning associated with motor-related processes (i.e., visual processing, visuospatial navigation, decision-making, imitation, and praxis) [28]. The right crus II is also associated with the default mode network, which is known to be affected in children with DCD [29]. Recently, our group found that after CO-OP intervention, there was increased default mode network activity, which supports an associated change in structure in right crus II. Furthermore, our group analyzed white matter microstructure and volumetric changes, using both VBM and DTI, following CO-OP intervention [30]. The results revealed increased white matter volume and/or fractional anisotropy in the posterior corona radiata, which contains axonal fibers of the corticospinal tract and plays a role in voluntary motor pathways in the brain [30]. Greater microstructural connectivity of this tract may improve the transmission of motor information from the cortex to the cerebellum and the spine, which is necessary for generating an internal model of movement [30]. These prior findings, along with our current findings, suggest that the CO-OP intervention has a joint effect on structure and function of the brain, specifically related to the cerebellum and associated networks.

Interestingly, no association between regional gray matter changes and BOT-2 was observed. This may be because CO-OP is not aimed at improving underlying motor skills; rather, it is task-specific [7]. Neuroplasticity theory states that specificity matters, meaning that the nature of the training experience dictates the nature of the plasticity. As such, it is unlikely an association between structural changes and underlying motor skills would be seen when our aim was to see task-specific changes related to CO-OP [31]. Subsequently, we investigated the association between PQRS and structural changes in the cerebellum. The PQRS measures movement quality for targeted child-chosen skills, rather than overall function. This allowed us to more readily see a robust positive association between the changes in regional gray matter volume and improvements in movement quality. Specifically, we found that improvements in PQRS-based movement quality predicted increases in gray matter volume in the motor (brainstem, and right and left lobule VIIIb), cognitive (right crus II), and emotional/self-regulation (right VIIb and vermis IX) regions of the cerebellum [28]. Sangster Jokić and Whitebread [26] found that DCD children exhibited ineffective self-regulatory behaviour during motor tasks. Our current results suggest that children with DCD may be able to better regulate themselves and their emotions using the CO-OP strategies. These improvements may assist a child in monitoring their performance, problem solving, and achieving their motor goals, which are also reflected through neuroplasticity in cerebellar motor and emotional/self-regulation regions.

The results of this study were limited by several factors. The sample size of our study was smaller than anticipated. This reduced power and prevented a planned comparison between children with DCD, and children with co-occurring DCD and ADHD. Most of our sample had clinically significant ADHD symptomology regardless of diagnosis, so the sample as a whole may be more similar than different, allowing us to combine the groups. In addition, limited sample size also prevented the treatment vs. waitlist comparison; therefore, our study is a pre-post study design. Given this context, it is possible that other, more subtle differences in gray matter exist in DCD, but we lacked the power to detect them. Restricted sample size was partially due to a high participant exclusion rate due to insufficient scan quality, which otherwise would have hampered the fidelity of the volumetric segmentation and increased the likelihood of type 1 error.

## 5. Conclusions

The current findings suggest that CO-OP is an effective intervention that improves motor function, movement quality, and motor skills and triggers salutary cerebellar plasticity in children with DCD. We observed increased gray matter volume in regions previously implicated in childhood DCD, further indicating the efficacy of CO-OP. Our findings indicate that the task-specific and cognitive nature of the intervention can induce neuroplastic changes in the motor, cognitive, and affective regions of the cerebellum. This study provides neuroscientific evidence for the benefits of CO-OP intervention, further strengthening the desirability of CO-OP as a standard of care for children with DCD.

## Figures and Tables

**Figure 1 brainsci-12-00856-f001:**
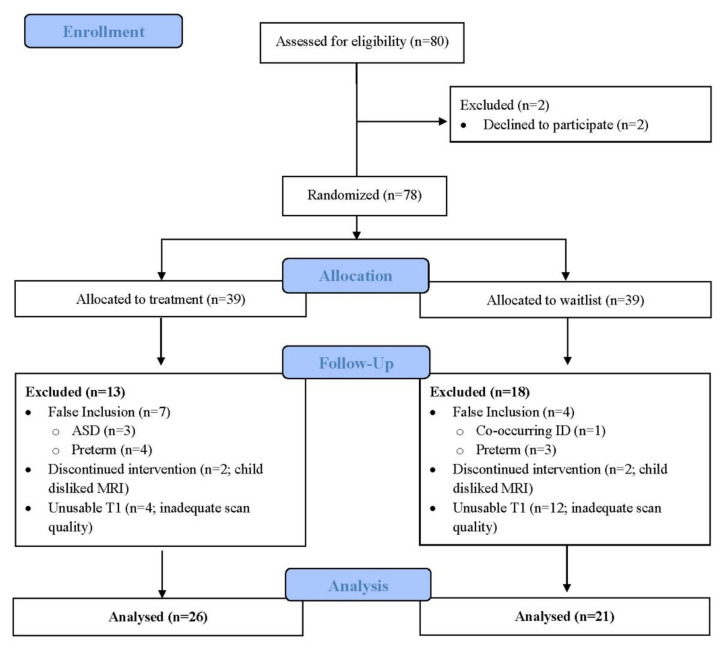
Participant flow diagram.

**Figure 2 brainsci-12-00856-f002:**
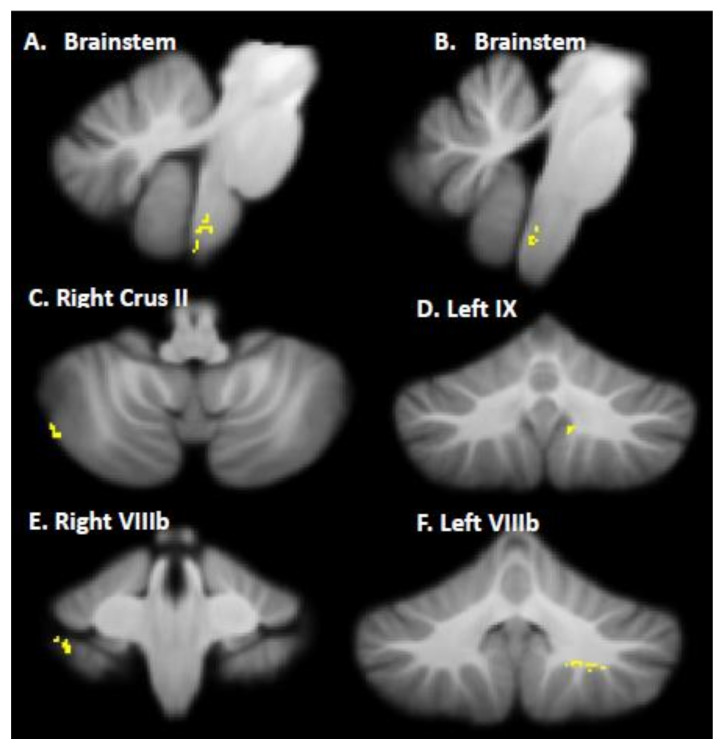
Increased gray matter volume in the brainstem, right crus II, left IX lobule, and bilateral VIIIb lobules after CO-OP intervention. Note: This figure corresponds to Table 3.

**Figure 3 brainsci-12-00856-f003:**
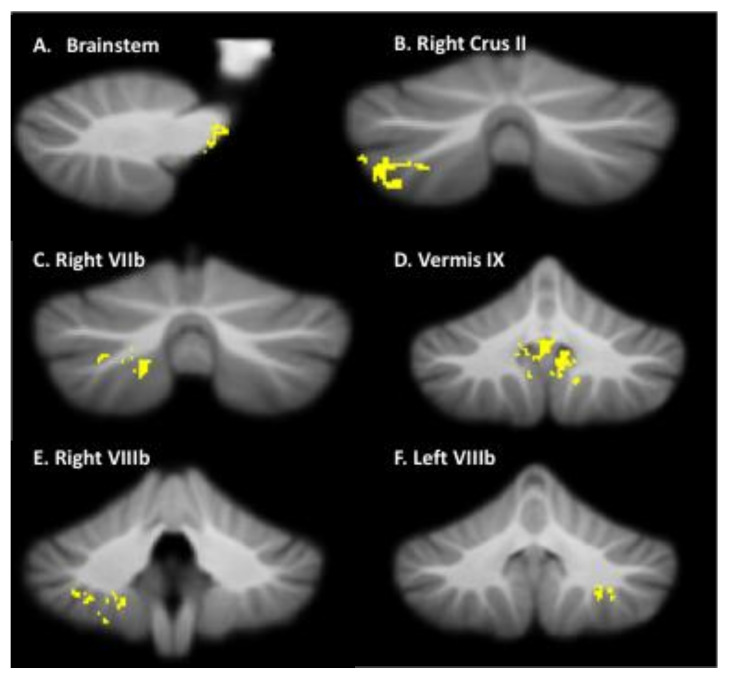
Regression analysis of Performance Quality Rating Scale (PQRS) and regional gray matter changes. This figure shows that higher PQRS scores predict increased gray matter volume in the brainstem, right crus II, right VIIb, vermis IX, and right and left VIIIb lobules. Note: This figure corresponds to Table 4.

**Table 1 brainsci-12-00856-t001:** Participant characteristics (*n* = 47).

Participant Characteristics	N (%) or Mean (SD)
Male	37 (79)
Age (years)	10.4 (1.44)
Total inter-cranial volume (L)	1.51 (0.16)
Conners 3 ADHD Index (*t*-score)	84 (11.2)
DCDQ	30.0 (9.40)
Total MABC-2 (percentile)	4.9 (7.91)

DCDQ, Developmental Coordination Disorder Questionnaire; MABC-2, Movement Assessment Battery for Children (2nd edition).

**Table 2 brainsci-12-00856-t002:** Outcomes before and after CO-OP intervention.

Measure	Pre-Test Mean (SD)	Post-Test Mean (SD)	*p*
COPM Performance	2.7 (1.4)	6.8 (1.3)	<0.001
COPM Satisfaction	2.7 (1.6)	7.5 (1.5)	<0.001
PQRS	3.1 (1.3)	6.0 (1.4)	<0.001
BOT-2 percentile	16 (14)	22 (19)	<0.001

BOT-2, Bruininks–Oseretsky Test of Motor Proficiency—2nd edition; COPM, Canadian Occupational. Performance Measure; PQRS, Performance Quality Rating Scale; SD, standard deviation.

**Table 3 brainsci-12-00856-t003:** MNI coordinates for significant gray matter volume increase following CO-OP intervention in children with DCD.

Location	X	Y	Z	*t*	Cluster Size (*k*)
Brainstem (A)	−6	−43	−55	2.43	63
Brainstem (B)	5	−44	−55	2.50	79
Right Crus II	49	−68	−46	3.30	27
Left IX	−9	−58	−42	2.39	29
Right VIIIa	38	−39	−45	2.89	28
Left VIIIa	−22	−53	−45	3.21	67

Note: This table refers to the clusters presented in Figure 2.

**Table 4 brainsci-12-00856-t004:** MNI coordinates for significant positive association between Performance Quality Rating Scale (PQRS) scores and gray matter volume following CO-OP intervention in children with DCD.

Location	X	Y	Z	*t*	*p*	Cluster Size (*k*)	Cohen’s d
Brainstem	−10	−26	−2	3.10	0.03	94	0.81
Right Crus II	40	−70	−50	3.28	0.005	539	0.57
Right VIIb	39	−63	−58	3.37	0.004	89	0.54
Vermis IX	−1	−55	−38	3.79	0.002	1075	0.61
Right VIIlb	25	−46	−51	3.34	0.004	356	0.67
Left VIIIb	−27	−49	−45	4.30	0.0001	214	0.51

Note: This table refers to the clusters presented in Figure 3.

## Data Availability

The data presented in this study are available from the corresponding author upon request. The data are not publicly available as the authors do not have REB approval to distribute them.

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
