# Peer review of "Cerebellar Differences after Rehabilitation in Children with Developmental Coordination Disorder"

_brainsci, 2022, doi:10.3390/brainsci12070856_

Round 1
Reviewer 1 Report
Thank you for your effort in this interesting field of work. And my compliments for the thoroughness of your study and the fully elaborated article. Please, to further improve the work, make the results section a bit more 'calm'. In my believe, many experts in the field of DCD could use a bit more of a step by step explanation in this type of research to really understand the results. And second, I would suggest a more broad conclusion in the sense that intervention in general effects motorskill and the increase of gray matter volume. Although it is the method used, the article should not become a CO-OP selling show.
Author Response
We thank reviewer 1 for their positive comments on the thoroughness of our study and fully elaborated article. We are unclear what the reviewer is requesting to make the results section a bit more “calm” but we have added some additional content in the results section and sub-sections that we hope addresses the reviewer’s concern. While we appreciate the reviewer’s suggestion to make the conclusion broader, we are concerned that the suggested change of “intervention in general effects motor skill and the increase of gray matter volume” goes beyond our data. We have made some edits to the conclusion to highlight the task-specific and cognitive nature of the intervention, but the emphasis is still on CO-OP as that is the only data we have presented. We cannot claim that other interventions will have the same effect on cerebellar neuroplasticity.
Reviewer 2 Report
The topic of this article is of great interest to many readers to understand the etiology of DCD and developmental problems. The authors' assertions can be clearly understood. Aadditions and supplements to the following points are considered necessary.
1. Why is it necessary to allocate the participants to two groups in the methodology of this study? One is the group intervened with CO-OP and the other is the control group? Please explain in detail in the text and in the legend in Figure 1.
2. Regarding Figure 1, the exclusion criterion "Not meeting inclusion criteria (n=0)" is mentioned, but if the n=0, I think there is no need to mention it.
3. In Result 3.1., a mention should be made in the text to refer to Table 1 here.
4. In Figure 2, the arrangements of A and B need to be changed.
5. The legends in Figures 2 and 3 refer to 'Note: This figure corresponds to Table...', which may be an error for 'Table 3' and 'Table 4' respectively. Please check.
Author Response
We thank reviewer 2 for their positive review and suggestions to improve the paper.
-
Thank you for bringing this to our attention. Since this study is a part of a larger study, participants were allocated to either the treatment group or the waitlist group for analysis using other neuroimaging modalities. Structural scans were taken from a larger data set that included various neuroimaging modalities, research questions, and analysis techniques. However, in this study, we have done a pre-post design in order to have sufficient power. We have provided more details in lines 100 and 104-108 on how we combined the scans of each group to create a “pre-intervention” group and a “post-intervention” group. We have corrected the error in Figure 1 to indicate the number allocated to the waitlist group.
-
Thank you for the suggestion. We have now amended Figure 1 and removed this extraneous information.
-
We had made reference to Table 1 in section 3.1. Please refer to line 188.
-
Thank you for catching our error. The figure has been adjusted accordingly.
-
Thank you for noticing this oversight on our part. We have now made the changes and are reflected on lines 234 and 251.
Reviewer 3 Report
This is a novel and very important new study. The paper is clearly written and the study is rigorously conducted.
I note the occupational therapist was blinded when scoring COPM data. Was the radiographer blinded to group allocation? This is worth noting.
Author Response
We appreciate the positive feedback from Reviewer 3.
Thank you for asking this question. The MRI technicians are hired independent of the study and were not aware of group allocation. This important consideration is now added to lines 108-109.